# Salt marshes create more extensive channel networks than mangroves

Christian Schwarz [1,2,3,4 ✉], Floris van Rees [4,5], Danghan Xie [4], Maarten G. Kleinhans [4] & Barend van Maanen [6]

Coastal wetlands fulfil important functions for biodiversity conservation and coastal protection, which are inextricably linked to typical morphological features like tidal channels. Channel network configurations in turn are shaped by bio-geomorphological feedbacks between vegetation, hydrodynamics and sediment transport. This study investigates the impact of two starkly different recruitment strategies between mangroves (fast/homogenous) and salt marshes (slow/patchy) on channel network properties. We first compare channel networks found in salt marshes and mangroves around the world and then demonstrate how observed channel patterns can be explained by vegetation establishment strategies using controlled experimental conditions. We find that salt marshes are dissected by more extensive channel networks and have shorter over-marsh flow paths than mangrove systems, while their branching patterns remain similar. This finding is supported by our laboratory experiments, which reveal that different recruitment strategies of mangroves and salt marshes hamper or facilitate channel development, respectively. Insights of our study are crucial to understand wetland resilience with rising sea-levels especially under climate-driven ecotone shifts.

[1] Department of Civil Engineering, KU Leuven, Leuven, Belgium. [2] Department of Earth and Environmental Sciences, KU Leuven, Leuven, Belgium. [3] School of Marine Science and Policy, University of Delaware, Lewes, DE, USA. [4] Department of Physical Geography, Utrecht University, Utrecht, The Netherlands. [5] Deltares, Rotterdamseweg 185, 2629 HD Delft, The Netherlands. [6] College of Life and Environmental Sciences, University of Exeter, Exeter, UK. ✉email: christian.schwarz@kuleuven.be

Mangroves and salt marshes occupy the margin between land and sea throughout a variety of geomorphological settings around the globe including river deltas, estuaries, coastal lagoons and open coasts[1] (Fig. 1). These coastal wetlands are amongst the most valuable ecosystems on the planet, providing a wealth of services to terrestrial and marine environments, and to society[2,3]. They function as biodiversity hotspots, store large amounts of carbon and play a crucial role in the protection against natural hazards as they shield the coast from waves and storms[4–6].

Salient features of coastal wetlands are tidal channel networks carving through the vegetated surface[7]. They connect the wetland with its adjacent water body and thereby (1) control the exchange of water, sediment and nutrients between land and sea[8] and (2) determine the spatial distribution of ecological niches and thus plant and benthic communities across the wetland[9,10]. Channel networks as a consequence govern coastal wetland evolution and resilience against external pressures, such as increased sea level rise[11,12]. Moreover, the network properties exert a decisive role in the propagation and attenuation of hydrodynamic energy and thus the degree to which wetland vegetation enhances coastal protection[13,14].

Within the last decades it has been established that vegetation actively shapes the formation of channel networks through the interaction between aboveground plant structures, tidal currents and sediment transport[15,16]. However, it was only discovered recently that plant growth strategies are important determinants for intertidal morphologies and channel network characteristics[11,17]. Implications of these findings on wetlands inhabited by mangroves or salt marsh plants around the globe remain still unknown. Species inhabiting both wetland types have the ability (1) to slow down tidal currents within vegetation, enhancing sedimentation and plant growth and (2) to promote flow acceleration and erosion in adjacent unvegetated areas hampering plant growth and initiating channel formation[16,18,19].

Despite these shared generic feedbacks, observations from satellite imagery indicate striking differences in channel networks (abundance, size and extent) between mangroves and salt marshes across different systems[16,20,21] (Fig. 1).

We hypothesize that different network properties arise from differences in colonization strategies. Most mangroves are known to expand as homogenous vegetation bands[22,23] caused by episodic recruitment events (i.e. windows of opportunities), during which they are able to establish a high amount of seedlings in a short amount of time (~weeks) through their viviparous germination strategy[24]. Vivipary is a common reproductive strategy across mangrove species (Supplementary Fig. 1), whereby seeds germinate on the parent tree and are detached and dispersed after developing into seedlings (e.g. *Avicennia sp.*, *Rhizophora sp.*)[25]. This reduces seedling susceptibility to disturbances, increases survival and results in a dense homogenous spatial vegetation cover[23,26–28]. In comparison to mangroves, salt marsh recruitment takes place more continuously through seeds, broken-off rhizome fragments and clonal expansion of existing patches[29–31]. The majority of temperate salt marshes host sub-species of the genus *Spartina* as their primary colonizers, which are characterized by relative low establishment probabilities from seeds but high rates of lateral clonal expansion leading to patchy vegetation cover during colonization[32–34] (Supplementary Fig. 2). In comparison to homogeneous mangrove coverage, such a patchy vegetation configuration is likely to cause distinct flow and sedimentation patterns and enhance channel formation[15,34–37]. Although literature also refers to patches in mangrove environments, mangrove patch sizes (ha) resulting from episodic establishment are several orders of magnitude larger than those of salt marsh patches (m) and therefore, in respect to channel formation with a characteristic length scale of meters, considered as homogenous[38].

From a local plant-flow interaction point of view, mangroves and salt marshes have comparable physical plant properties.

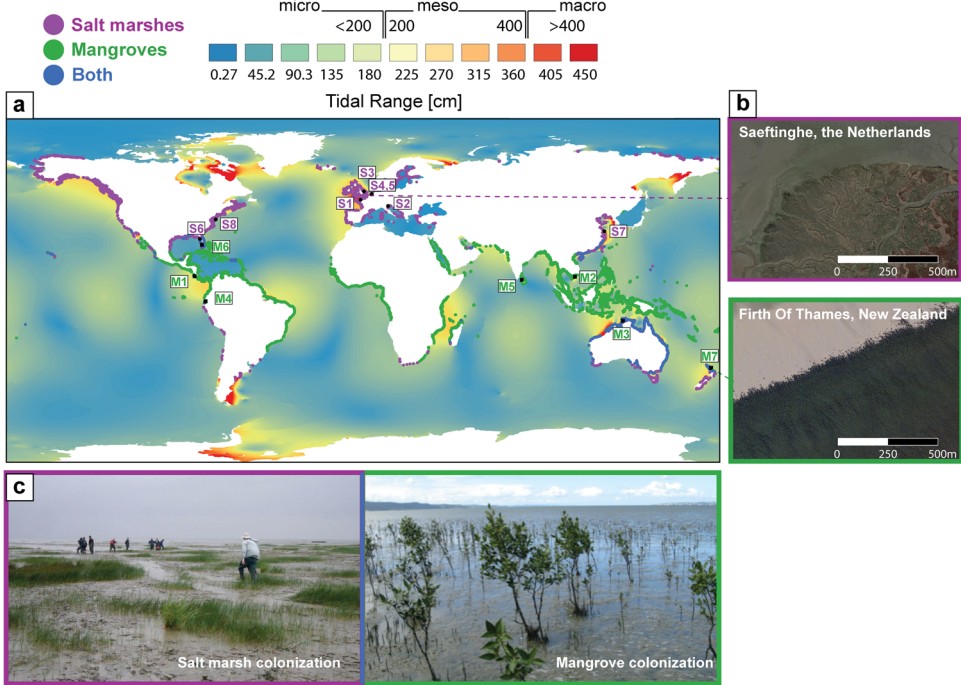

**Fig. 1 Study sites. a** Global distribution of salt marshes (purple), mangroves (green) and coexistence of both (blue) relative to the M2-tidal range[20, 21], S1-8 and M1-7 are salt marsh and mangrove systems investigated in this study; **b** example of distinct channel networks in salt marsh and mangrove systems. **c** Snapshot of different colonization strategies between salt marsh and mangrove systems. Salt marshes colonize slowly (~years[76]) creating a patchy spatial pattern, whereas mangroves colonize fast (~weeks[24]) creating a homogenous spatial pattern.

Plants of both systems produce a high density of more or less cylindrical elements (aerial roots and stems for mangroves, plant stems for salt marshes) slowing down flow and facilitating sediment depostion[39–41]. Our hypothesis that vegetation colonization impacts channel networks inherently assumes that the initial network structure (created during vegetation establishment) remains imprinted in the landscape while the system matures. This is in agreement with previous studies showing that although channels continue to elaborate after initial incision, the presence of vegetation and its precipitated increase in bed strength and reduction in flow velocity impedes the development of new channels and leaves the majority of the channel network unaltered[42–44].

This study investigates the hypothesis that differences in channel networks between salt marshes and mangroves originate from different spatio-temporal colonization patterns through (1) a remote sensing based channel network comparison of mangroves and salt marshes around the globe, and (2) a scaled lab experiment investigating the ramifications of different colonization behaviors on channel development. We use three channel network metrics to compare channel networks; (1) the Hortonian drainage density (D), which gives a simple description of the degree of channelization but does not distinguish different patterns of network branching[45]; (2) the mean unchanneled path length (mUpl), a measure of the distance a drop of water, placed on the vegetated platform would need to travel to reach the closest channel, therefore indicating how efficient tidal channels drain a watershed[45,46]; (3) geometric efficiency ($\frac{l_H}{mUpl}$), the Hortonian length ($l_H$, the inverse of the Hortonian drainage density, $D^{-1}$) divided by the mean unchanneled path length (mUpl), which characterizes channel geometric patterns like branching and meandering of the channel network[45,47]. We find that salt marshes create more extensive and effective networks than mangroves, while the channel geometry and branching patterns remain similar. The difference in channel networks is found to be controlled by the variations in bio-geomorphological interactions arising from different colonization patterns.

## Results

**Network characteristics**. A global comparison of watershed drainage densities between 8 salt marsh and 7 mangrove systems with varying tidal ranges reveals significant differences in channel network characteristics. The average salt marsh watershed drainage density ($2.49 \times 10^{-2} \, \mathrm{m}^{-1}$) was about 6 times the average mangrove watershed drainage density ($0.41 \times 10^{-2} \, \mathrm{m}^{-1}$) (p value: 0.015) (Fig. 2a–c). This difference is consistent with the comparison in the mean drainage density for entire wetlands between the two wetland types (p value: 0.02) (Fig. 2c). Correspondingly, a comparison between mean unchanneled path lengths reveals significantly larger mean path lengths (3x) for mangroves in comparison to salt marsh systems (p value: 0.014) (Fig. 2d–f). Geometric efficiency does not exhibit significant differences between the two wetland types (Fig. 2g–i).

A comparison between the total channel length, the sum of all channel lengths, as a function of the watershed area follows the previously described tidal range independent power law relationship found for salt marshes in Great Britain, Italy and the US[42,45,48] (Fig. 3). The increased slope (b) for salt marshes compared to mangroves could hint to a faster increase in channel length to drainage area. However the 5 green mangrove points on the left side of the graph (Area: $10^3 – 10^4$) driving the difference in this relationship belong to a very small mangrove system (Whitianga) which might offset the otherwise spatially constant scaling of network development[45], or suggest that power law scaling is less pronounced at mangrove systems (Fig. 3). An

ANOVA-test comparing the regression models for mangrove and salt marshes suggests the models differ significantly (p value: 0.0034)

**Scaled lab experiment**. Our flume experiments show clear differences in morphological development between the homogenous and patchy vegetation configuration representing mangroves and salt marshes, respectively. On bare surfaces, the size of self-formed channels depends on flow and sediment properties, but on partly vegetated surfaces the scale of nonuniform flow resistance may affect channel dimensions and branching pattern. The homogenous configuration inhibits channel formation, while the patchy configuration promotes it (Fig. 4a). This becomes especially visible when comparing the number of channels along the longitudinal axis of the flume (Fig. 4b) and the drainage density (Fig. 4c), showing that vegetation patches accelerate channel network formation, which results in higher overall channel abundance and causes channels to extend and enlarge further into the vegetated surface. At the seaward boundary of the vegetated zone, where flood conditions are the same and there is more bare surface scape to form more parallel channels in the patchy experiment, a lower number of channels in fact formed in the patchy experiment, that were also larger than in the homogenous experiment because of the higher drainage density and further landward extension of channels. This is caused by the flow concentration and concurrent increased flow momentum in the patchy experiment.

## Discussion

Our remote sensing analysis shows the emergence of distinct drainage densities and mean unchanneled path lengths distinguishing salt marsh and mangrove systems, while geometric efficiencies remain similar. Differences in drainage densities and mean unchanneled paths suggest that salt marshes host more extensive and effective channel networks (higher D and lower mUpl) than mangrove systems. However, the similarity in geometric efficiency suggests similar channel geometries and branching patterns between the two wetland types (Fig. 2). Thus, networks between the two systems, although geometrically similar, exhibit clear differences in channel abundance across different tidal range conditions (Figs. 2, 3). We therefore suggest that saltmarsh channels create an increased exchange of water, sediment and nutrients between the vegetated surface and the adjacent water body per unit time compared to mangrove channels. Our selection of real systems covers micro-, meso- and macrotidal conditions and collectively encapsulates a broad range of coastal wetland systems (Supplementary Table S1, Supplementary Figs. 4, 5), suggesting that channel extent between mangrove and salt marsh systems differs from each other. Our scaled flume experiments show that colonization patterns can exert a major control on channel network properties, with patchy vegetation configurations resulting in a higher drainage density than the homogeneous configurations. While previous research has suggested that network geometry (i.e. geometric efficiency) is shaped by plant-flow interactions and resulting bio-geomorphological feedbacks[45,47], here we show that colonization strategies determine the overall degree of wetland channelization. We interpret our results in respect to interactions between time scales of vegetation pattern development and morphodynamics. If the time scale of vegetation colonization is slower or equal to morphodynamical adaptations (i.e. salt marsh case), plant-flow interactions create patchy vegetation patterns and promote channel incision. If vegetation colonization is faster than morphodynamical adaptations (i.e. mangrove case), plant-flow interactions are inhibited leading to homogeneous vegetation cover in which

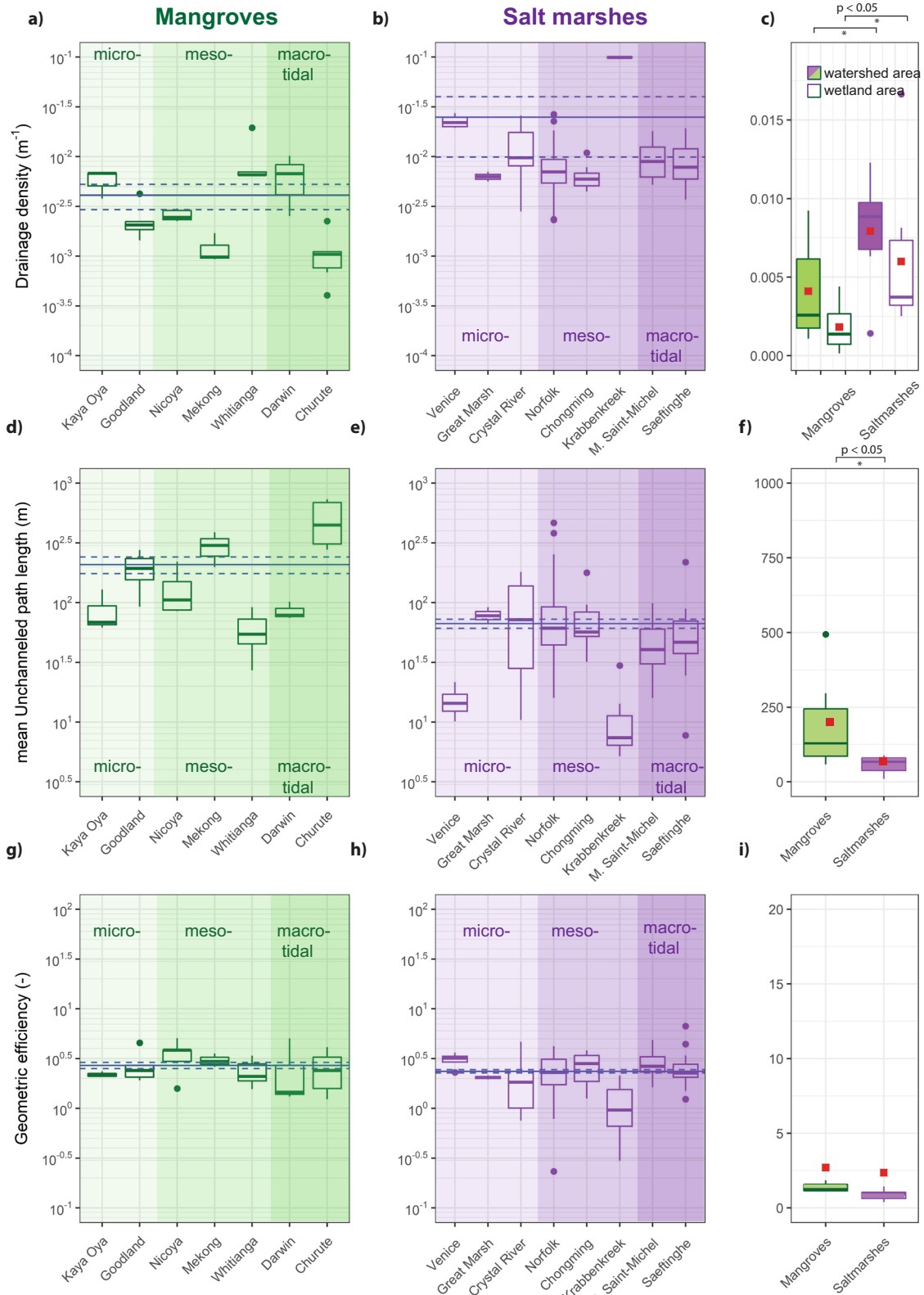

**Fig. 2 Channel metric comparisons between mangrove and salt marsh systems.** Comparison of drainage density (D), mean unchanneled path length (mUpl) and geometric efficiency ($l_H/mUpl$) between mangroves and salt marshes; **a**, **d**, **g** and **b**, **e**, **h** present above proxies for each system per delineated watershed, with mean values marked by solid-blue lines, and the standard error from the mean value marked by blue-dashed lines; **c**, **f**, **i** All proxies averaged per location and grouped between mangrove and salt marsh environments, red dots constitute average values, the white box-plots in (**c**) represent the drainage density based on the entire wetland areas. Statistically significant differences are indicated with an asterisk (*), based on two-sided two sample Wilcoxon-tests. Tidal range did not exert a significant impact on network proxies (Supplementary Fig. 3).

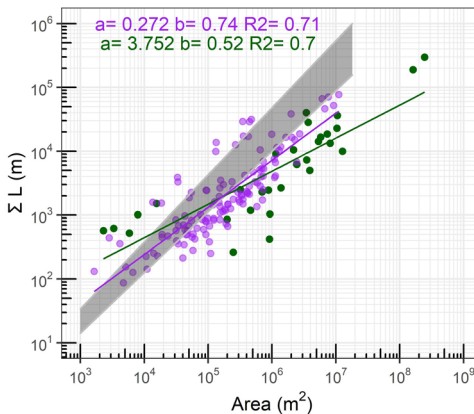

**Fig. 3 Channel scaling for mangrove and salt marsh systems.** Total network length and watershed area for mangrove and salt marsh watersheds showing a common power law relationship ($\sum L = aArea^b$), for which salt marshes indicate more extensive channel networks at comparable drainage area sizes compared to mangroves. Marani et al. (2003) found a similar relation for sub-basins in Venetian salt marshes which is plotted as a grey area ($\sum L = 0.020 \times Area^{1\pm0.5}$). Salt marsh linear model-fit: F-statistic: 311.6 on 1 and 126 DF, p-value < 2.2 $e^{-16}$, Mangrove linear model fit, F-statistic: 66.39 on 1 and 29 DF, p-value < 5.52 $e^{-9}$; More information on the influence of the correction factor on this relationship can be found in Supplementary Fig. 9d, e.

reduced flow velocities and increased bed strength hinder channel incision.

The remote sensing approach is limited by the spatial resolution of the used remote sensing products and therefore potentially underrepresents channels around and below a width of 3 m. Techniques extracting channel networks with higher resolution generally rely on bathymetric data collected during LIDAR surveys[48]. Nevertheless, high-resolution data requirements prevent a comparison of systems around the globe using the same data source. Also, field studies on mangrove systems using in-situ observations and LIDAR, support the observed scarcity of channels in mangrove systems[14,49]. Given that our approach omits the identification of smaller channels which are predominantly present in salt marsh systems, more accurate remote sensing approaches would likely reveal even larger differences in drainage density and mean unchanneled path length between salt marsh and mangrove wetlands.

The formation of tidal channels is an inherently complex process dependent on the coastal setting (e.g. coastal slope, substrate type, sediment erodibility, and existing mudflat channels), and hydrodynamic characteristics (e.g. tidal prism, -asymmetry and wave activity) as well as bio-geomorphological feedbacks complicating the interpretation of channel forming mechanisms[50,51]. Previous studies distinguishing an initial channel incision phase from a subsequent channel evolution phase already hinted that the network configuration is defined during the initial channel incision phase[19,44]. This underpins the importance of plant colonization strategies in determining channel network properties as found by our experiment. However, the observed variation among mangrove systems, with the Mekong delta (Vietnam) and the ecological reserve of Churute (Ecuador) characterized by an almost complete absence of tidal channels, while mangrove environments in the Darwin Harbour (Australia) host extensive tidal channels, still poses questions on the role of environmental settings (Fig. 2a and Supplementary Fig. 4)[52,9]. More specifically, under which circumstances are channel patterns driven by coastal geometry and large-scale morphodynamics (i.e. Darwin Harbour) dominating over channel patterns driven by vegetation colonization strategies and plant-flow interactions (i.e. Mekong and Churute)? Are plant colonization patterns mainly important on local scales, while geological constraints and tidal flow patterns determine channels on the system, e.g. inlet or estuary scale? These questions provide important directions for future studies.

Our results underline that bio-geomorphological feedbacks acting on the watershed scale are able to form vastly different channel networks, with some scatter in network properties caused by tidal range and relative elevation (Supplementary Table 1, Fig. 3). Although the simplification of fast-homogenous and slow-patchy colonization strategies for mangroves and salt marshes does not do justice to the variety of growth characteristics[11,22], our comparison suggests that evolutionary selection in colonization strategies, in particular vivipary, might have major implications on coastal morphologies and potentially coastal resilience[53]. Channel networks have a primary control on the distribution of sediment and hydrodynamic energy across the vegetated surface[13]. More specifically, the presence of more extensive and potentially deeper tidal channel networks created by plant-flow interactions allows sediment to reach areas further inland[54] and alters inorganic sedimentation on the vegetated platform[55]. For mangrove systems, where tidal channels are more limited, sediment is predominantly deposited in the more seaward zone of the forest leaving the landward side devoid of sediment input[27,56]. This can cause sediment starvation in the upper forest so that sea-level rise is more likely to outpace sediment accretion rates, with potential negative impacts on mangrove cover and diversity[40]. Thus, channel network properties are potentially important determinants of whether entire wetlands or merely subsections can keep up with rising sea levels, which has important implications on estimated carbon budgets and projected carbon sequestration rates, which has not been explored yet[57–59]. Channel network properties also affect how effectively coastal vegetation dampens storm surges as the presence of low-drag channels allows flood tidal waves to propagate much further inland, reducing their coastal protection function[14]. Consequently, climate-driven ecotone shifts, might not only change species diversity of salt marshes and mangroves but also channel dynamics, and thus resilience and services provided by coastal wetlands.

## Methods

**Image processing and network analysis**. We extracted tidal channel networks from multi-spectral satellite images to be able to consistently (using the same data source and techniques) compare systems with different tidal ranges at characteristic geomorphological settings (open coast, lagoon and estuary) around the globe. Our analysis utilized globally available multi-spectral (R, G, B, NIR) high-resolution satellite images (pixel size of 3 x 3 m) from Planet Labs (www.planet.com), with a constellation of 150–200 nano-satellites "Doves" collecting imagery at 3–5 m resolution on a daily scale. To obtain a representative overview of global wetland characteristics, fifteen sites were selected varying in tidal range (micro-, meso- and macro-tidal) and coastal wetland type (salt marsh and mangrove) (Fig. 1). Image selection per site was based on a cloud cover <5% and the most recent acquisition date of the 4-band PlanetScope scene (PS4scene, 3 m spatial resolution) image product. An overview of the investigated sites and their characteristics can be found in Supplementary Table 1.

PS4scene image products are based on spectral bands and bandwidths of red (R) with a 590–670 nm range, green (G) with a 500–590 nm range, blue (B) with a 455–515 nm range and near-infrared (NIR) with a 780–860 nm range. Images with a small positional error (<10 m RMSE) were selected and georeferenced in WGS 1984 UTM coordinate system within their corresponding zone. We used the top of atmosphere radiance (TOA) 16-bit unsigned digital number pixel values as data source. A conversion to radiance or reflectance was omitted since the classification was based on relative differences in vegetation and water-related indices and extracted channel networks, which were then compared between images.

Each wetland system was defined following its land and sea boundaries marking the border of the area of interest (AOI). For each watershed within an investigated wetland system, we quantified the Hortonian drainage density, the mean unchanneled path length and geometric efficiency as proxies for channel network characteristics. To compare these metrics between wetland types, we averaged each

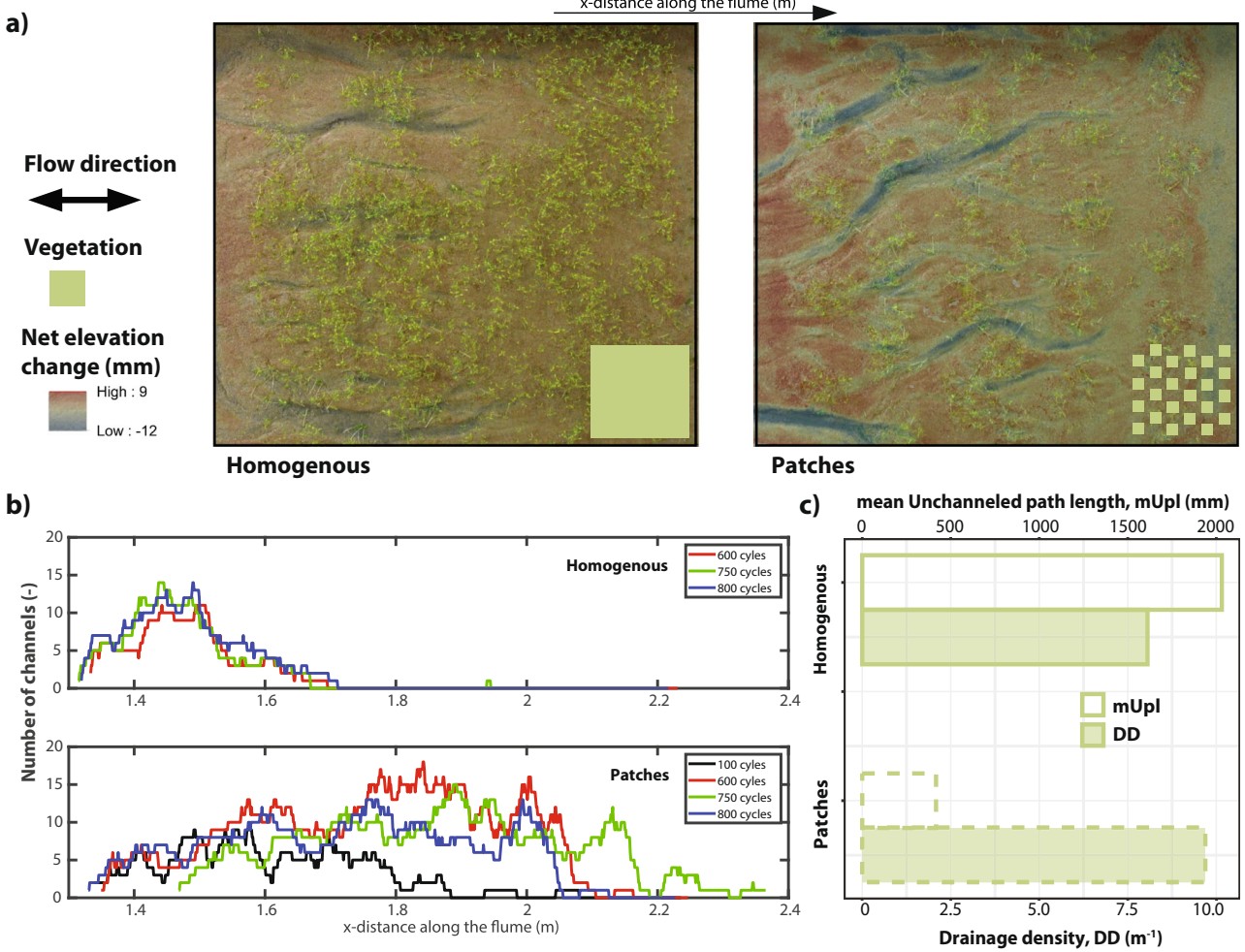

**Fig. 4 Channel development in the laboratory experiment. a** Sedimentation-erosion map of the vegetation scenarios; **b** number of channels over time (cycles represent time-scale of channel development) and distance along the flume; **c** drainage densities and mean unchanneled path length of the homogenous and patchy vegetation configuration.

metric per location to create an equal distribution of mean values per wetland type (Fig. 2c, f, i). The Hortonian drainage density, $DD = \frac{\sum L}{A} = \frac{1}{l_H}$, is the ratio between the total channelized network length ($L$) and the watershed area ($A$) and thus gives a simple description of the degree of channelization, though it fails to distinguish geometric or topological characteristics[60]. To exclude artefacts related to delineation of watersheds (Supplementary Fig. 8) and to be able to evaluate the extent of channels in respect to the entire wetland we moreover calculated the drainage density based on the entire wetland area, $DD_{wetland} = \frac{\sum L}{A_{wetland}}$ (Fig. 2c). The mean unchanneled path length, $mUpl$, in tidal systems is defined as the mean shortest distance a parcel of water, placed on the vegetated surface, would have to travel across the platform before reaching the nearest channel. As such, it is a measure of how efficiently the network drains or feeds the wetland[40,45,61]. The geometric efficiency, is the Hortonian length ($l_H$, which is the inverse of the Hortonian drainage density, $DD^{-1}$) divided by the mean unchanneled path length (i.e. $\frac{l_H}{mUpl}$). It is as a metric that indicates the efficiency with which, for a certain Hortonian drainage density, the wetland is dissected by the channel network and thus characterizes channel geometric patterns like branching and meandering[47,62]. Channel network analysis based on satellite data can be complicated by the limits in available spatial resolution (Planet Labs cell-size: 3 m) and by above-ground biomass (e.g. salt marsh grasses or mangrove crowns) of wetland vegetation obscuring small (low Hortonian order) channels or creeks. Therefore, we carried out a sensitivity analysis, establishing correction factors to account for these inaccuracies based on the following steps. (1) We compared channel networks extracted from satellite data with channel networks extracted from high-resolution digital elevation models for one mangrove (Whitianga, New Zealand)[63] and one salt marsh system (Saeftinghe, the Netherlands) using the same extraction procedure (Rijkswaterstaat, RWS.nl). This comparison resulted in visible differences between satellite and DEM based channel metrics. (2) To correct for these differences in our global data set we successively removed low order channels (1st–5th order) of the DEM data and compared it's extracted channel metrics with

the satellite data (Supplementary Fig. 9a, b). (3) We then established an exponential relationship between removed channel order and channel metrics for mangroves and salt marshes, which was used to correct the satellite based metrics (Supplementary Fig. 9c). Due to the scarcity of high resolution elevation data, correction factors were based on only one system per wetland type.

Statistical testing was carried out using the R software package[64]. For comparing channel metrics between salt marsh and mangrove systems, we used a Shapiro-Wilk test to test for normality of our datasets. Since our datasets where non-normally distributed, statistical testing was carried out using a two-sided two samples Wilcox Rank test.

Watershed areas were delineated using a euclidean distance algorithm in ArcGis 10.5. Channel networks were extracted and converted to polygons using an unsupervised maximum likelihood classifier (ten classes) available in ArcGis 10.5 and the free version of XtoolsPro (Supplementary Figs. 6–8)[65]. If necessary, classes were manually reclassified as wetland or channel based on two criteria: 1) vegetation is bordered by a visible/ embankment on one side and bordered by a large water body (e.g. estuary, sea, bay) on the other side, and 2) vegetation is bordered by a sudden change in vegetation class on one side and bordered by the sea on the other side. Channel lengths were calculated by measuring centerlines of the created channel polygons[66]. For further details on the remote sensing analysis please refer to Supplementary Figs. 6–8. The allometric scaling relationship between the total channel lengths ($y$; m) and watershed areas ($x$; m²) is described by a power function, $y = ax^b$, where $a$ and $b$ are fitted constants. We used ordinary least squares fitting on the log-transformed powerfunction $\log y = \log a + b \log x$ to compare scaling relations between mangroves and salt marshes[48]. To compare the difference in resulting regression models for salt marshes and mangroves, we used an anova function between the complete dataset and the complete dataset weighed by the wetland type.

**Lab experiment setup and data analysis.** Previous studies have shown that physical scale models can be valuable additions to field measurements in understanding system dynamics in a controlled/simplified tidal environment[67,68].

**Table 1 Relevant hydrodynamic parameters in nature in comparison with hydrodynamic parameters in the experiment.**

|  | Velocity ($U$) [m/s] | Period ($T$) [s] | Length ($L$) [m] | $\frac{U \times T}{L}$[-] | Mobility [-] |
|---|---|---|---|---|---|
| Experiment ($e$) | 0.05–0.4 | 81.1–112.5 | 3.5 | 1.2–12.2 | 0.05–3.7 |
| Nature ($n$) | 0.05–0.3 | 44712 | 26.6–4503.3 (620) | 3.6–21.7 | 0.25–2 |

Here, $U_e$, observed flow velocity values for the experiment were based on measurements during bare runs of this study[18]; $U_n$, flow velocity values for natural systems are based on salt marsh observations[18]; $T_e$, the tidal period for the physical scale model (i.e. periods of reduced asymmetric tilt and asymmetric tilt), $T_n$, the tidal period for nature is the period of the M2 tide; $L_e$, length of the experimental flow length, $L_n$, symbolizes the length-scale represented by the Hortonian length ($D^{-1}$) of the investigated wetlands (range and mean value); $(U \times T)/L$ is the hydrodynamic scaling parameter. We compare sediment mobility between nature and the experiment by comparing the mobility parameter, which we define as the ratio between shear stresses and critical shear stress observed in nature (BSS: 0.1–0.8[73, 74] N m$^{-2}$; BSS$_{crit}$ 0.4[78] N m$^{-2}$) and in our experiment (BSS: 0.03–1.8[73, 74] N m$^{-2}$; BSS$_{crit}$ 0.5[75] N m$^{-2}$).

However, it has been proven difficult to study morphological development by tidal forcing using these physical scale models[69]. The main issue is to overcome general scaling rules associated with tidal flow[70], as experiments may produce low sediment mobility, unrealistically deep scour holes and ripples associated with hydraulic smooth conditions[68,69]. However, more recently successful experiments were conducted in flumes with a tilting bed. The periodical tilting of the entire flume prevents scale-related issues and induces similar sediment mobility in the ebb and flood direction as observed in 'real-scale' field situations[69,71]. Results have shown that careful selection of sediment composition contributes to a better representation of processes, therefore, a sand mixture with $D_{50}$ of 0.55 mm, a $D_{10}$ of 0.32 mm and a $D_{90}$ of 1.2 mm was chosen[71]. To ensure similar hydrodynamic conditions between scaled experiment and field conditions we followed previously proposed scaling relationships. More specifically, the flow velocity and tidal period scaled by the characteristic length have been kept constant between the physical scale model and reality, see Eq. 1[67,72].

$$\frac{U_e \times T_e}{L_e} = \frac{U_n \times T_n}{L_n} \quad (1)$$

Here, $U_{e,n}$ is the flow velocity, $T_{e,n}$ is the period of the tide, $L_{e,n}$ is the characteristic length, subscript $e,n$ signifies the values for the scale experiment and natural field conditions, respectively (Table 1).

Previous research identified that the distribution of excess momentum around vegetation objects is crucial for channelization[19]. As such, in addition to morphological and hydrodynamic scaling, vegetation was scaled based on previous scale experiments[72].

The flume experiment was carried out in a 3.5 m × 1.2 m tilting flume with a 0.95 m vegetated area at its center and bordered by an open water body at one side and a dissipation ramp with a 0.125 m/m slope at the other (Supplementary Fig. 10). The tilt is controlled by a mechanical jack that moves up and down and was connected to the ground. The entire flume pivots around a central axis of the flume, with a fixed maximum and minimum tilting amplitude, tilting speed and a delay at the maximum and minimum tilting moments (Supplementary Fig. 10). We used *Medicago sativa* to represent vegetation objects and tested two vegetation configurations (homogeneous and patchy) planted with equal seedling densities. The experiments were developed to primarily focus on the principle of channel formation with two different types of vegetation growth patterns. As such, we decided to use the same plant species for both experimental runs. A homogeneous configuration was realized with seeds planted over the complete vegetation area while a patchy configuration was constructed with interpatch distances of 10 cm and 5 cm in the length and width of the flume, respectively (Supplementary Fig. 11). It is important to note that the vegetation patterns for both mangroves and salt marshes are static relative to changes in morphology, thus plants do not die-off or grow throughout the experiment. This inherently assumes that in the scaled experiments mangrove development is much faster than the morphodynamic time scale, and salt marsh development is much slower than the morphodynamic timescale[11,24].

Changes in morphology were recorded by three overhead photo cameras. Pictures were taken in phase with the mechanical jack supported by software programs Photoboot and PSRemote to capture minimum tilt amplitude, maximum tilt amplitude and two zero crossings during rising tide and falling tide. These four images were taken after each 10th tidal cycle. Moreover, a stereo system scanner and additional software (VX Studio) were used to generate DEMs. Every DEM consisted of seven individual images of approximately 50 cm × 90 cm. These images were corrected for anomalies of the scanner and converted into DEM using structure for motion in the MATLAB R2017a software package. For more details on the experimental setup and choices on morphological and vegetation scaling please refer to Supplementary Figs. 10–11 and Table 3.

## Data availability

The data generated in this study have been deposited in the Zenodo database under accession code (https://doi.org/10.5281/zenodo.6331067).

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

## Acknowledgements

We thank Erik M. Horstman for providing digital elevation data of the mangrove site Whitianga, New Zealand used for ground-truthing of our analysis. We thank B. Evans, A. D'Alpaos and two anonymous reviewers and the editor for their constructive feedback.

## Author contributions

C.S. and b.v.M. designed this project. C.S., B.v.M., M.G.K. and F.v.R. designed the study approach. F.v.R. and D.X. conducted the experiments and data analysis and carried out the remote sensing analysis. F.v.R. conducted and analyzed the scaled flume experiment. C.S., B.v.M., F.v.R., D.X. and M.G.K. wrote the paper. All authors provided comments on the data processing and substantially contributed to the drafts.

## Competing interests

The authors declare no competing interests.
