## [Peer Review File · Nature Communications]

Salt marshes create more extensive channel networks than mangrovesReviewers' Comments:

Reviewer #1:

Remarks to the Author:

The paper presents an analysis of channel network structure in saltmarshes and mangrove systems and concludes that saltmarshes develop a more extensive channel network than mangroves due to their patchiness. The ideas behind the paper are novel and of potential interest of the research community.

The authors use mainly two sources of evidence to support their statement. The first one is based on the extraction of channel networks using satellite multi-spectral information on 12 study sites, corresponding to six mangrove wetlands and six saltmarsh wetlands. The second one is based on the extraction of channel networks in two sets of laboratory experiment, one with homogeneous vegetation cover and the other one with a regular patch vegetation arrangement.

While the rationale and the design of the research seems in principle well suited to study the research question, I believe that there are some methodological problems that need consideration.

1. The extraction of the channel network in the 12 study sites has been done by combining spectral information that identifies vegetation with information that identifies water areas. The method of extraction is not verified against any independent network data, so we cannot assess its accuracy and, more importantly, we cannot assess if the method works with a similar degree of accuracy in saltmarsh (grasses) and mangroves (trees). Characteristics of both vegetation types are quite different, so water detection classification may have different challenges in both settings. Without a validation of the network extraction method in both vegetation types we cannot fully assess if the differences in drainage density is due to the type of vegetation or if it is due to systematic errors in the network extraction procedures. It would not be unreasonable to expect an underestimation of channels in mangrove wetlands that tend to have a dense canopy. That could change the results presented in figure 3.

2. Most of the saltmarsh sites are in Europe, which limits the variety of environments analyzed in the study.

3. The laboratory experiments need more explanation regarding the channel initiation process, and they may not be completely representative of the saltmarsh-mangrove differences. They really represent two types of saltmarsh, one with uniform coverage and the other with patches. It could be argued that with a sparser coverage of vegetation (patches) more channels will form, so the results are not surprising. The configuration with patches will have more areas with low erosion resistance, so more channelization will occur.

The resistance to flow and erosion will be most likely different in saltmarshes and in mangroves, so spatial organization may not be the only factor that results in differences in channel network density.

Reviewer #2:

Remarks to the Author:

In this manuscript, the authors investigate differences in the characteristics of tidal channel networks in salt-marsh and mangrove systems. Tidal channel networks are important morphological features of tidal systems for their control of the ecomorphodynamic evolution of those systems. The issue addressed in the paper is therefore relevant and timely, given current concerns on coastal system response to changes in the environmental forcing. Indeed, analyzing and possibly predicting how different vegetation types might affect channel-network form and function is a key step to further improve current knowledge of the biological functioning of tidal ecosystems. This study proposes the idea that the different vegetation recruitment strategies in mangroves and salt marshes control channel network properties. Based on remote-sensing and experimental analyses the authors suggest that tidal networks in salt-marsh systems are more extensive and effective than in mangrove systems. They interpret this result as a consequence of the different colonization patterns in mangroves and salt-marsh systems that control tidal-network properties.

The idea has merit as it combines observations from actual network structures to the analysis of laboratory-generated networks, and the issue is quite timely and relevant, as I said above. I have however a few observations and suggestions that need to be addressed before the paper is accepted for publication. Let me reinforce, however, that I feel that the paper could significantly add to the literature on this important issue.

1) An important issue concerns the considered stage of network development. The authors suggest that vegetation recruitment strategies in mangroves and salt marshes control channel network incision and properties. This is suggested to occur at the initial stages of network development. The authors further assume that the first network structure remains imprinted in the landscape, an observation put forth by many in the literature as acknowledged by the authors. However, besides this first network imprinting, long-term network evolution dynamics through branching and meandering might influence network properties and structure (see Kearney and Fagherazzi, 2016). And indeed, what the authors analyze in this study through remote sensing analyses, is network structure at later stages of the evolution. How can the authors prove that the initial imprinting dictated by vegetation growth strategies controls network form and function at later stages?

2) Also, the variability in the three considered metrics (Hortonian drainage density, mean unchanneled path length, and geometric efficiency) at the system scale is much larger than the average difference in the same metric between mangrove and marsh systems. How might this influence the robustness of the results?

3) When considering actual networks and synthetic channel patterns, the authors consider different metrics. Why do they do that? Why don't they consider the above recalled three metrics also for the laboratory-generated networks? Why don't they consider the number of channels in actual network cases?

4) What other physical and biological features might control network properties? Excluding the tidal range that has been considered by the authors, how do the elevation of the vegetated platforms, sediment cohesion, sediment size, vegetation density at later stages control network geometry? Are platform elevations within the tidal frame different in the two cases? This might control channel length and size.

Given the above-recalled interactions, how do the authors prove that the differences are only due to colonization strategies and that other factors do not play any role?

5) What about networks developed over unvegetated landscapes? Should they have a lower or a higher drainage density? Might the unvegetated/vegetated marsh ratio considered by Ganju et al. (2017) be a relevant quantity to analyze?

Detailed comments

Line 25. Tidal channels are some of the typical morphological features in tidal systems. I suggest changing to "linked to typical morphological features like tidal channels".

Line 30. I suggest changing to "vegetation establishment strategies"

Line 31. Over salt marshes, not only do tidal channel networks display a larger total length, but they also dissect the marsh by decreasing the unchanneled length. This should be recalled by modifying the sentence "We find that salt marshes are dissected by more extensive channel networks", maybe adding something to highlight differences in network efficiency.

Line 32. No quantitative descriptions are provided on branching and meandering structures and patterns (no analysis of branching frequency and/or channel sinuosity). I, therefore, suggest deleting or modify this sentence.

Lines 51-52. It should be clarified, to avoid misunderstanding, that no consistent relationships have been found between vegetation distribution and distance to channels. The zonation patterns observed in marshes mostly depend on marsh elevation (e.g. Marai et al., 2013) which can be considered as a

proxy for the distribution of other edaphic conditions that control the spatial distribution of vegetation communities.

Lines 80-83. However, van Maanen et al. (2015) highlighted differences in network features for different tidal ranges in mangrove settings, and van Maanen et al. (2015) highlighted differences in network features for different tidal ranges in tidal embayments. Are these lines consistent with the results of the above modeling frameworks? Please clarify.

Line 197. "Cause" should be "causes"

Lines 234-239 I think these lines deserve a more detailed discussion. I am not convinced that "the time scale of vegetation development is slower or equal to morphodynamical adaptations (i.e. salt marsh case)". Marshes tend to accrete vertically at rates that mimic sea level rise, i.e. a few mm/yr. Can we safely say that the time scale of vegetation adaptation is slower than marsh adaptation (at least in the vertical frame)? If we talk about network first imprinting, I agree that it occurs over shorter temporal scales. But usually, marsh vegetation is adapted to marsh elevation (and vice-versa given the well-known biogeomorphic feedbacks).

Reviewer #3:

Remarks to the Author:

I enjoyed reading the manuscript in which the authors hypothesize that reproductive traits and the seedling distributions in which they result can lead to contrasts in large-scale channel morphologies between mangrove and salt marsh systems. This is an interesting hypothesis and well worth exploring. The implications of these contrasts in channel geometries for the systems themselves and society as a whole are clear, and of concern in the current context of global change. The work opens up a lot of interesting questions to explore in future studies. The study is therefore timely, novel and of broad significance. I found the manuscript well written and clearly structured with figures and data appropriate to the question. I have a few concerns over some of the assumptions underlying the analyses presented, however, which I feel should be addressed before the paper is suitable for publication. These are detailed as major points below:

Major comments:

1. It strikes me that there are hints within the data presented (especially within-class in Fig 2, and a possible non-linearity in supplementary fig 3) that tidal range may indeed exert a control on channel morphology. I therefore question the authors' claims of independence from tidal range. It appears to me that, with small numbers of samples ($n=6$), and a sampling strategy that was not stratified with the primary aim of investigating effects of tidal range, the lack of a significant relationship does not imply independence, particularly given the hints within the data that a relationship may well emerge if the sample size were larger or experimental design aimed at detecting this relationship. I accept that the tidal range effect may be secondary to the vegetation effect, but not that it doesn't play a role. I would therefore like to see the authors adjust the semantics around their claims and discussions as regards tidal range and vegetation to recognise this fact.

2. Sample size is quite small ($n=6$) to argue representativeness at global scale, and there is no detail provided as to how the sites were selected, so it is difficult to tell whether the results reflect a selection bias in choice of sites, or a genuine signal that would be evident if any six randomly selected marshes/mangroves were selected. It seems to me that it would be easy to (unconsciously) select twelve locations that fit the hypothesis, and that the sampling strategy must therefore explicitly guard against this. Can the authors comment on how their sites were selected and the way in which this selection was independent of their existing knowledge/assessment of the channel morphologies?

3. As the authors touch upon in their discussion, there is an important relationship here between the resolution at which channels are detected and the resulting network descriptor values (L240-248). A 3m resolution image does not necessarily imply that a 3m wide channel will be detectable. Typically features become detectable only well above the resolution of the imagery – or at least the confidence

with which they can be detected declines towards the pixel scale as pixels become mixed and uncertain. I would therefore guess that the actual detection limit for channels in this analysis is probably above five metres for saltmarsh, while the larger, taller canopies of mangroves may obscure channel surfaces and overhang banks such that the realistic detectable width is actually significantly larger than that for salt marshes. Without ground validation data, which are not presented, it is therefore difficult to judge whether the assumption that is implicit in the analysis, that the networks in the two systems are delineated with the same spatial resolution, and are therefore comparable, is justified (even though the method may be the same). It seems likely, based on the different growth forms of the two systems, that smaller channels are already under-represented in the mangrove dataset, which may account for some of the difference observed that is attributed to vegetation colonisation strategies. The authors should either provide ground validation data to show that this measurement bias is negligible and their extracted networks are comparable, or explore the sensitivity of their overall findings to this potential (uncontrolled) bias through a sensitivity analysis. How different would the channel detection scales need to be between the two systems for the contrasts found in the analysis to be lost? Is it reasonable to think that whatever that difference is has not been exceeded in this study, and that the findings may therefore be interpreted as robust? On a further, and related, point, the authors use the most recent available satellite image (L295), without regard for tidal stage at time of capture. Depending on which indices are used as features for the classifier, which are not specified but are stated to be vegetation- and water-based (303-304), whether channels are filled with water or drained will probably have a strong influence on the contrast in index values seen at the channel boundary, and therefore the confidence with which channels of a particular scale can be delineated. Similarly, if the classification is dominated by water-related indices, then a half-full channel would probably result in its width being estimated as that of the water surface, not the channel itself. It would therefore seem important to control for tidal stage in the image selection also, or account for the potential uncertainty introduced in the analysis and discussion.

Minor comments:

Caption to Fig. 1, and in text: Can the terms "slowly" and "fast" have indicative contextual timescales applied to them? Is there evidence that can be cited to support this contrast in a less abstract way?

L82: 'Seems decoupled from tidal range' – these references don't support this point explicitly. Are there references that do, or is this decoupling something that this study needs to explicitly demonstrate itself, in which case this sentence will need rewriting to reflect. This touches on my point 2 above.

L94: 'Primary colonizers' – what evidence is there for this ubiquity? There needs to be a reference to support this statement, and I don't think those provided really do. Without a robust justification for focusing on *Spartina* (because it is almost everywhere), with its well-documented patchy colonisation patterns, the findings of the study could be quite niche, and only applicable to a few locations. I don't question that *Spartina* and its growth forms are a reasonable basis for this study, but would like to see the global relevance better supported.

L95: 'lateral expansion' – do the authors mean 'clonal expansion'?

L103: some, but not all, plants in both systems produce a high density of cylinders.

L106: assumption that network structure imprinted and stable... OK, this is well documented, but what about things like headward creek incision and so forth that is likely to lead to the densest channel networks? What about marshes in degenerative states where the initial channel network starts to become lost to internal dissection processes, which will dramatically alter the channel metrics.

Figure 2: There appears to me to be evidence within classes that the channel metrics are sensitive to tidal range, even if the differences between classes exceed these. See point 1 above.

Supplementary Figure 3: The authors claim linear relationships, but it looks like one could just as easily fit non-linear functions to the points presented. Can the authors provide fit statistics for linear and non-linear fits to show that the data do indeed best fit a linear relationship? Also spelling of Length on X axis label.

Figure 4: Was the lab experiment only run once? If not, how many replicates, and can any statistical

test be offered to show that the differences are significant? Admittedly, the contrast shown is convincing if only one test was conducted, but must then be interpreted more explicitly in the context of a single observation and qualitative comparison. The colours in the top-down photos don't reproduce brilliantly – can they be altered to make the vegetation elements more distinct from the elevation change overlay?

L222: 'geometrically'

L256-262: Interesting questions that definitely need to be addressed. Is a major contrast between the sites also the sediment delivery rate?

L263: The spatial scale at which plant colonisation is important compared to geological constrains, and whether there is anything in between is interesting. It also becomes a temporal scale issue as older networks are likely to be larger (even if inherited from smaller, plant-driven geometries). Larger networks also more easily detected at a given resolution...

Do the authors get a sense of what they mean by 'local scales' and what the upper scale limit at which plant colonization effects are dominant might be? It seems possible that this upper limit might be different depending on the plant too. There are all sorts of interesting questions here!

L273: Please provide a reference for the 'deeper' argument.

L275: If channel planforms are more limited in mangroves, doesn't that imply that they will probably have to be more hydraulically efficient in order to exchange a given volume of sediment? This is obviously beyond the scope of this study, but doesn't hydraulic efficiency complicate the inferences made here about the implications of different networks for stability, surge attenuation and so forth?

L279: 'determinants OF whether'

L281: 'effectiveLY'

Ben Evans

REVIEWER COMMENTS

Reviewer #1:

The paper presents an analysis of channel network structure in saltmarshes and mangrove systems and concludes that saltmarshes develop a more extensive channel network than mangroves due to their patchiness. The ideas behind the paper are novel and of potential interest of the research community.

The authors use mainly two sources of evidence to support their statement. The first one is based on the extraction of channel networks using satellite multi-spectral information on 12 study sites, corresponding to six mangrove wetlands and six saltmarsh wetlands. The second one is based on the extraction of channel networks in two sets of laboratory experiment, one with homogeneous vegetation cover and the other one with a regular patch vegetation arrangement.

While the rationale and the design of the research seems in principle well suited to study the research question, I believe that there are some methodological problems that need consideration.

We want to thank the reviewer for the encouraging assessment.

1. The extraction of the channel network in the 12 study sites has been done by combining spectral information that identifies vegetation with information that identifies water areas. The method of extraction is not verified against any independent network data, so we cannot assess its accuracy and, more importantly, we cannot assess if the method works with a similar degree of accuracy in saltmarsh (grasses) and mangroves (trees). Characteristics of both vegetation types are quite different, so water detection classification may have different challenges in both settings. Without a validation of the network extraction method in both vegetation types we cannot fully assess if the differences in drainage density is due to the type of vegetation or if it is due to systematic errors in the network extraction procedures. It would not be unreasonable to expect an underestimation of channels in mangrove wetlands that tend to have a dense canopy. That could change the results presented in figure 3.

We want to thank the reviewer for pointing this out. We now provide a measure of the accuracy of our channel extraction method. (1) We compare our satellite data based extractions to high resolution DEMs to assess what type of channels (based on average width) are missing from our detection through a combination of plant-characteristics (e.g. canopy shading by mangrove trees) and image resolution. (2) We establish a statistical model between successive removal of higher order channels and extracted channel proxies and use it to determine a correction factor which we use on our dataset.

Line 337ff: "Channel network analysis based on satellite data can be complicated by above-ground biomass (e.g. salt marsh grasses or mangrove crowns) of wetland vegetation obscuring small (low hortonian order) channels or creeks. Therefore, we carried out a sensitivity analysis, establishing correction factors to account for these inaccuracies. We (1) compare satellite-based channel networks with high-resolution digital elevation models for one mangrove (Whitianga, New Zealand) and one salt marsh system (Saeftinghe, the Netherlands) (Rijkswaterstaat, RWS.nl)⁶⁴. (2) This data is compared to channel metrics of DEM channel networks with successively removed low order channels (1st - 5th order). (3) Then we establish a relationship between removed channel order and channel metric, which is used to correct the satellite based metrics (Supplementary Fig. 9a,b)."

2. Most of the saltmarsh sites are in Europe, which limits the variety of environments analyzed in the study.

We addressed this comment by adding salt marsh sites in China and the United States (Fig.1,2 and 3). The extracted channel metrics fall within the observed range of the previously selected sites and thus underline the robustness of the trend which we already observed.

3. The laboratory experiments need more explanation regarding the channel initiation process, and they may not be completely representative of the saltmarsh-mangrove differences. They really represent two types of saltmarsh, one with uniform coverage and the other with patches. It could be argued that with a sparser coverage of vegetation (patches) more channels will form, so the results are not surprising. The configuration with patches will have more areas with low erosion resistance, so more channelization will occur. The resistance to flow and erosion will be most likely different in saltmarshes and in mangroves, so spatial organization may not be the only factor that results in differences in channel network density.

More explanation has now been added. Indeed, the experiments were developed to primarily focus on the principle of channel formation with two different types of vegetation growth patterns. As such, we decided to use the same plant species for both experimental runs. While we agree that the results are intuitive, the number of channels is in fact much larger within the patchy vegetation zone, in comparison to the adjacent bare area. Thus, channelization is not simply the inverse function of vegetation coverage, and patchy vegetation actively promotes channel formation. Also, channel initiation has its own intrinsic scales depending mainly on width, flow momentum and sediment mobility (Leuven et al. 2016). If these scales were disregarded, it could be argued that channels can form between individual stems rather than tussocks or patches. It could also be argued that twice the area of bare surface in the patchy experiment should result in twice the number of channels. However, even under the controlled laboratory conditions, the self-organisation of the channel pattern is more complex than that. The experiments show a certain size of channels with widths, depths and bends in mutual proportion and in relation to the flow momentum, meaning that a large number of possible flow paths that could have become channels did not become channels. Given the same settings of both experiments and the control on the vegetation pattern, it is relevant evidence for the main point of the paper that the channels differ in size and number at the edge of the vegetated zone in the middle of the flume (at x-coordinate 1.5 m) and throughout the vegetation zone.

We also make that clearer now in the manuscript:

Line 418: "The experiments were developed to primarily focus on the principle of channel formation with two different types of vegetation growth patterns. As such, we decided to use the same plant species for both experimental runs"

Leuven, J. R. F. W., Kleinhans, M. G., Weisscher, S. A. H., & Van der Vegt, M. (2016). Tidal sand bar dimensions and shapes in estuaries. *Earth-science reviews*, 161, 204-223.

Reviewer #2:

In this manuscript, the authors investigate differences in the characteristics of tidal channel networks in salt-marsh and mangrove systems. Tidal channel networks are important morphological features of tidal systems for their control of the geomorphological evolution of those systems. The issue addressed in the paper is therefore relevant and timely, given current concerns on coastal system response to changes in the environmental forcing. Indeed, analyzing and possibly predicting how different vegetation types might affect channel-network form and function is a key step to further improve current knowledge of the biological functioning of tidal ecosystems. This study proposes the idea that the different vegetation recruitment strategies in mangroves and salt marshes control channel network properties. Based on remote-sensing and experimental analyses the authors suggest that tidal networks in salt-marsh systems are more extensive and effective than in mangrove systems. They interpret this result as a consequence of the different colonization patterns in mangroves and salt-marsh systems that control tidal-network properties.

The idea has merit as it combines observations from actual network structures to the analysis of laboratory-generated networks, and the issue is quite timely and relevant, as I said above. I have however a few observations and suggestions that need to be addressed before the paper is accepted for publication. Let me reinforce, however, that I feel that the paper could significantly add to the literature on this important issue.

We want to thank the reviewer for the positive assessment. Below we address all the reviewer's remarks in a point by point manner.

1) An important issue concerns the considered stage of network development. The authors suggest that vegetation recruitment strategies in mangroves and salt marshes control channel network incision and properties. This is suggested to occur at the initial stages of network development. The authors further assume that the first network structure remains imprinted in the landscape, an observation put forth by many in the literature as acknowledged by the authors. However, besides this first network imprinting, long-term network evolution dynamics through branching and meandering might influence network properties and structure (see Kearney and Fagherazzi, 2016). And indeed, what the authors analyze in this study through remote sensing analyses, is network structure at later stages of the evolution. How can the authors prove that the initial imprinting dictated by vegetation growth strategies controls network form and function at later stages?

For natural systems, we cannot prove this without availability of historic imagery or geologic reconstruction. However, it is plausible, as meandering may occur in a subset of tidal systems but is not ubiquitous due to the high threshold of erosion of muddy and rooted sediments. In the experiments, meandering is not occurring either, but branching and stream capture are (see supplemented media). However, the flow concentration and concurrent increased flow momentum in the patchy experiment affects the channel pattern different from the more 'spatially diffuse' homogenous experiment and fundamentally changed it in that much larger channels penetrated much deeper into the vegetated zone of the patchy experiment.

Added in the text in Line 222ff:

“At the seaward boundary of the vegetated zone, where flood conditions are the same and there is more bare surface scape to form more parallel channels in the patchy experiment, but in fact a lower number of channels formed in the patchy experiment, that were also larger than in the homogenous experiment because of the higher drainage density and further landward extension of channels. This is caused by the flow concentration and concurrent increased flow momentum in the patchy experiment.”

Moreover, previous research has shown that, when the tidal landscape reaches an elevation that allows the colonization by halophytic vegetation, this freezes the configuration of the network which can, from then on, only undergo minor changes immaterial to its basic structure. Our hypothesis that vegetation colonization impacts channel networks inherently assumes that, although longer term network dynamics may influence network properties, the initial network structure (created during vegetation establishment) remains largely imprinted in the landscape while the system matures (Marani 2003, Taramelli 2018).

Marani, M., Belluco, E., D'Alpaos, A., Defina, A., Lanzoni, S., & Rinaldo, A. (2003). On the drainage density of tidal networks. *Water Resources Research*, 39(2).

Taramelli, A., Valentini, E., Cornacchia, L., Monbaliu, J., & Sabbe, K. (2018). Indications of dynamic effects on scaling relationships between channel sinuosity and vegetation patch size across a salt marsh platform. *Journal of Geophysical Research: Earth Surface*, 123(10), 2714-2731.

2) Also, the variability in the three considered metrics (Hortonian drainage density, mean unchanneled path length, and geometric efficiency) at the system scale is much larger than the average difference in the same metric between mangrove and marsh systems. How might this influence the robustness of the results?

We now provide a sensitivity analysis showing the impact of removal of different channel orders on mean unchanneled path length and drainage density. Supplementary Fig.9a,b.

3) When considering actual networks and synthetic channel patterns, the authors consider different metrics. Why do they do that? Why don't they consider the above recalled three metrics also for the laboratory-generated networks? Why don't they consider the number of channels in actual network cases?

We thank the reviewer for his suggestion. That we ask these questions is one of the great values of experimentation, but they are not easy to solve. Intuitively, channel depth scales with water depth, which in the intertidal zone is of course related to tidal amplitude. However a conceptually more valid idea is that the channel cross-sectional areas depend on the tidal prism upstream of each cross-section, which is to date still very difficult to determine because it requires accurate bathymetry as well as tidal divides between the source areas of each channel. However in our experiment the study/drainage areas are comparable which is

why we can compare the number of channels as a valid metric, which is not the case for our field sites.

We now added the same metrics in the lab-experiments as shown in the GIS-data analysis (Fig.2 and 4)

4) What other physical and biological features might control network properties? Excluding the tidal range that has been considered by the authors, how do the elevation of the vegetated platforms, sediment cohesion, sediment size, vegetation density at later stages control network geometry? Are platform elevations within the tidal frame different in the two cases? This might control channel length and size.

Given the above-recalled interactions, how do the authors prove that the differences are only due to colonization strategies and that other factors do not play any role?

Variations in sediment composition, cohesion and in vegetation density may well account for the scatter observed in our analysis as indicated by studies focusing on the salt marsh-system scale (e.g. Schwarz et al. 2018), but assessing this in the present data would require much (expensive) field data that is unavailable. However for inter-system comparison the most important factors, as side from vegetation, to explain network characteristics were shown to be tidal range and rel. elevation in respect to mean sea level (Liu et al. 2021). The latter we now added in the supplementary information(Supplementary Table.1) and discuss in text:

Line 295: “Our results underline that bio-geomorphological feedbacks acting on the watershed scale are able to form vastly different channel networks, with some scatter network properties caused by . tidal range and relative elevation (Supplementary Table 1, Fig.3).”

Liu, Z., Gourgue, O., & Fagherazzi, S. (2021). Biotic and abiotic factors control the geomorphic characteristics of channel networks in salt marshes. *Limnology and Oceanography*.

Schwarz, C., Gourgue, O., Van Belzen, J., Zhu, Z., Bouma, T. J., Van De Koppel, J., ... & Temmerman, S. (2018). Self-organization of a biogeomorphic landscape controlled by plant life-history traits. *Nature Geoscience*, 11(9), 672-677.

5) What about networks developed over unvegetated landscapes? Should they have a lower or a higher drainage density? Might the unvegetated/vegetated marsh ratio considered by Ganju et al. (2017) be a relevant quantity to analyze?

The reviewer poses an interesting question. Unfortunately this question is outside of the scope of the current study, especially keeping in mind that Kearney and Fagherazzi, (2016), showed the high variety in unvegetated channel network structures which potentially is linked to sediment characteristics and the relatively higher sediment mobility in the absence of vegetation. Moreover it is still unclear how unvegetated landscapes would compare with mangrove environments. Indeed the UVVR, unvegetated/vegetated marsh ratios, has been shown to be robust proxy scaling with sediment budgets of microtidal (and more recently meso and macro-tidal) marsh complexes and therefore a reliable proxy for marsh

vulnerability. However since this metric was previously only applied to salt marsh systems a validation for mangroves would warrant a detailed investigation of sediment budgets, which is beyond the scope of the current study.

Detailed comments

Line 25. Tidal channels are some of the typical morphological features in tidal systems. I suggest changing to “linked to typical morphological features like tidal channels”.

Changed as suggested

Line 30. I suggest changing to “vegetation establishment strategies”

Changed as suggested

Line 31. Over salt marshes, not only do tidal channel networks display a larger total length, but they also dissect the marsh by decreasing the unchanneled length. This should be recalled by modifying the sentence “We find that salt marshes are dissected by more extensive channel networks”, maybe adding something to highlight differences in network efficiency.

Changed into:

Line 33:” We find that salt marshes are dissected by more extensive channel networks and characterized by shorter over-marsh flow paths than mangrove systems, while branching patterns remain similar.”

Line 32. No quantitative descriptions are provided on branching and meandering structures and patterns (no analysis of branching frequency and/or channel sinuosity). I, therefore, suggest deleting or modify this sentence.

We thank the reviewer for this suggestion. According to Marani 2003, the network efficiency also gives an indication of the branching structure; therefore we keep this statement in the sentence, but removed the meandering-statement.

Lines 51-52. It should be clarified, to avoid misunderstanding, that no consistent relationships have been found between vegetation distribution and distance to channels. The zonation patterns observed in marshes mostly depend on marsh elevation (e.g. Marai et al., 2013) which can be considered as a proxy for the distribution of other edaphic conditions that control the spatial distribution of vegetation communities.

Following studies of for instance Sanderson et al. 2000, Taramelli 2018 or D’Alpaos 2016 we think there is a (at least indirect) relationship present in literature between channel networks and vegetation zonation. Although, as mentioned by the reviewer, there are indications that co-varying marsh elevation or increased subsoil aeration control these relationships the resulting pattern is nevertheless robust which is why we kept the sentence as is.

D’Alpaos, A., & Marani, M. (2016). Reading the signatures of biologic–geomorphic feedbacks in salt-marsh landscapes. *Advances in water resources*, 93, 265-275.

Sanderson, Eric W., Susan L. Ustin, and Theodore C. Foin. "The influence of tidal channels on the distribution of salt marsh plant species in Petaluma Marsh, CA, USA." *Plant Ecology* 146.1 (2000): 29-41.

Taramelli, A., Valentini, E., Cornacchia, L., Monbaliu, J., & Sabbe, K. (2018). Indications of dynamic effects on scaling relationships between channel sinuosity and vegetation patch size across a salt marsh platform. *Journal of Geophysical Research: Earth Surface*, 123(10), 2714-2731.

Lines 80-83. However, van Maanen et al. (2015) highlighted differences in network features for different tidal ranges in mangrove settings, and van Maanen et al. (2015) highlighted differences in network features for different tidal ranges in tidal embayments. Are these lines consistent with the results of the above modeling frameworks? Please clarify.

We thank the reviewer for this suggestion. Van Maanen 2015, uses a constant tidal range of 2m in his model study when assessing channel formation in mangrove environments, however this reference is definitely relevant for the contextualization of our study that we accidentally omitted and now added. However, van Maanen 2013 explored different tidal ranges but without vegetation.

Line 91 :” Despite these shared generic feedbacks, observations from satellite imagery indicate striking differences in channel networks (abundance, size and extent) between mangroves and salt marshes across different systems, which seem decoupled from variations in main forcing factors such as tidal range ^{16,20,21} (Fig. 1). “”

van Maanen, B., Coco, G., & Bryan, K. R. (2015). On the ecogeomorphological feedbacks that control tidal channel network evolution in a sandy mangrove setting. *Proceedings of the Royal Society A: Mathematical, Physical and Engineering Sciences*, 471(2180), 20150115.

van Maanen, B., Coco, G., & Bryan, K. R. (2013). Modelling the effects of tidal range and initial bathymetry on the morphological evolution of tidal embayments. *Geomorphology*, 191, 23-34.

Line 197. “Cause” should be “causes”

Changed as suggested

Lines 234-239 I think these lines deserve a more detailed discussion. I am not convinced that “the time scale of vegetation development is slower or equal to morphodynamical adaptations (i.e. salt marsh case)”. Marshes tend to accrete vertically at rates that mimic sea level rise, i.e. a few mm/yr. Can we safely say that the time scale of vegetation adaptation is slower than marsh adaptation (at least in the vertical frame)? If we talk about network first imprinting, I agree that it occurs over shorter temporal scales. But usually, marsh vegetation is adapted to marsh elevation (and vice-versa given the well-known biogeomorphic feedbacks).

We thank the reviewer for pointing this out. We now understand that the phrasing was misleading. We agree with the reviewer that we cannot safely say that vegetation adaptation

is slower than marsh adaptation (at least in the vertical frame). However, our point focuses on the first imprinting of the tidal network. More specifically, since mangroves colonize and form closed meadow in a matter of weeks, bio-geomorphological feedbacks potentially leading to tidal channels cannot emerge as effectively.

We are now clarifying this point in Line264:

“If the time scale of vegetation colonization is slower or equal to morphodynamical adaptations (i.e. salt marsh case), plant-flow interactions create patchy vegetation patterns and promote channel incision. If vegetation colonization is faster than morphodynamical adaptations (i.e. mangrove case), plant-flow interactions are inhibited leading to homogeneous vegetation cover in which reduced flow velocities and increased bed strength hinder channel incision.”

Reviewer #3 Ben Evans:

I enjoyed reading the manuscript in which the authors hypothesize that reproductive traits and the seedling distributions in which they result can lead to contrasts in large-scale channel morphologies between mangrove and salt marsh systems. This is an interesting hypothesis and well worth exploring. The implications of these contrasts in channel geometries for the systems themselves and society as a whole are clear, and of concern in the current context of global change. The work opens up a lot of interesting questions to explore in future studies. The study is therefore timely, novel and of broad significance. I found the manuscript well written and clearly structured with figures and data appropriate to the question. I have a few concerns over some of the assumptions underlying the analyses presented, however, which I feel should be addressed before the paper is suitable for publication.

We want to thank the reviewer for the encouraging assessment. Below we address all the reviewer's remarks in a point by point manner.

These are detailed as major points below:

Major comments:

1. It strikes me that there are hints within the data presented (especially within-class in Fig 2, and a possible non-linearity in supplementary fig 3) that tidal range may indeed exert a control on channel morphology. I therefore question the authors' claims of independence from tidal range. It appears to me that, with small numbers of samples ($n=6$), and a sampling strategy that was not stratified with the primary aim of investigating effects of tidal range, the lack of a significant relationship does not imply independence, particularly given the hints within the data that a relationship may well emerge if the sample size were larger or experimental design aimed at detecting this relationship. I accept that the tidal range effect may be secondary to the vegetation effect, but not that it doesn't play a role. I would therefore like to see the authors adjust the semantics around their claims and discussions as regards tidal range and vegetation to recognise this fact.

We agree with the reviewer's assessment. In particular, our main message being that the difference in channel networks is primarily caused by vegetation colonization strategies. However, we nevertheless would expect that within colonization strategies an impact of tidal range might exist.

We now make that more clear in Line: 295:

"Our results underline that bio-geomorphological feedbacks acting on the watershed scale are able to form vastly different channel networks, with some scatter in network properties caused by tidal range and relative elevation (Supplementary Table 1, Fig.3)."

2. Sample size is quite small ($n=6$) to argue representativeness at global scale, and there is no detail provided as to how the sites were selected, so it is difficult to tell whether the results reflect a selection bias in choice of sites, or a genuine signal that would be evident if any six randomly selected marshes/mangroves were selected. It seems to me that it would be easy to (unconsciously) select twelve locations that fit the hypothesis, and that the

sampling strategy must therefore explicitly guard against this. Can the authors comment on how their sites were selected and the way in which this selection was independent of their existing knowledge/assessment of the channel morphologies?

We thank the reviewer for the opportunity to elaborate on our sampling design. We selected 2 replicates per tidal range, wetland type and characteristic geomorphological settings. Geomorphological settings considered are Estuaries, Lagoons and Open coast environments, which is not a selection of sites that could fit the hypothesis, but a selection made for a broad range of fundamentally different sites to put the hypothesis to a test. We now show these details in the Supplementary Table.1 and elaborate on this in adjusted text:

Line 318: “We extracted tidal channel networks from multi-spectral satellite images to be able to consistently (using the same data source and techniques) compare systems with different tidal ranges at characteristic geomorphological settings (open coast, lagoon and estuary) around the globe.”

3. As the authors touch upon in their discussion, there is an important relationship here between the resolution at which channels are detected and the resulting network descriptor values (L240-248). A 3m resolution image does not necessarily imply that a 3m wide channel will be detectable. Typically features become detectable only well above the resolution of the imagery – or at least the confidence with which they can be detected declines towards the pixel scale as pixels become mixed and uncertain. I would therefore guess that the actual detection limit for channels in this analysis is probably above five metres for saltmarsh, while the larger, taller canopies of mangroves may obscure channel surfaces and overhang banks such that the realistic detectable width is actually significantly larger than that for salt marshes. Without ground validation data, which are not presented, it is therefore difficult to judge whether the assumption that is implicit in the analysis, that the networks in the two systems are delineated with the same spatial resolution, and are therefore comparable, is justified (even though the method may be the same). It seems likely, based on the different growth forms of the two systems, that smaller channels are already under-represented in the mangrove dataset, which may account for some of the difference observed that is attributed to vegetation colonisation strategies. The authors should either provide ground validation data to show that this measurement bias is negligible and their extracted networks are comparable, or explore the sensitivity of their overall findings to this potential (uncontrolled) bias through a sensitivity analysis. How different would the channel detection scales need to be between the two systems for the contrasts found in the analysis to be lost? Is it reasonable to think that whatever that difference is has not been exceeded in this study, and that the findings may therefore be interpreted as robust? On a further, and related, point, the authors use the most recent available satellite image (L295), without regard for tidal stage at time of capture. Depending on which indices are used as features for the classifier, which are not specified but are stated to be vegetation- and water-based (303-304), whether channels are filled with water or drained will probably have a strong influence on the contrast in index values seen at the channel boundary, and therefore the confidence with which channels of a particular scale can be delineated. Similarly, if the classification is dominated by water-related indices, then a half-full channel would probably result in its width being estimated as that of the water surface,

not the channel itself. It would therefore seem important to control for tidal stage in the image selection also, or account for the potential uncertainty introduced in the analysis and discussion.

We thank the reviewer for this suggestion and now address the problem of data resolution and potential shading by aboveground plant structures.

We now provide a measure of the accuracy of our channel extraction method. (1) We compare our satellite data based extractions to high resolution DEMs to assess what type of channels (based on average width) are missing from our detection through a combination of plant-characteristics (e.g. canopy shading by mangrove trees) and image resolution. (2) We use a statistical model between successive removal of higher order channels and extracted channel proxies and use it to determine a correction factor which we use on our dataset.

Line 353ff: "Channel network analysis based on satellite data can be complicated by above-ground biomass (e.g. salt marsh grasses or mangrove crowns) of wetland vegetation obscuring small (low hortonian order) channels or creeks. Therefore, we carried out a sensitivity analysis, establishing correction factors to account for these inaccuracies. We (1) compare satellite-based channel networks with high-resolution digital elevation models for one mangrove (Whitianga, New Zealand) and one salt marsh system (Saeftinghe, the Netherlands) (Rijkswaterstaat, RWS.nl)⁶⁴. (2) This data is compared to channel metrics of DEM channel networks with successively removed low order channels (1st - 5th order). (3) Then we establish a relationship between removed channel order and channel metric, which is used to correct the satellite based metrics (Supplementary Fig. 9a,b)."

Minor comments:

Caption to Fig. 1, and in text: Can the terms "slowly" and "fast" have indicative contextual timescales applied to them? Is there evidence that can be cited to support this contrast in a less abstract way?

We thank the reviewer for this suggestion and now added the respective time-scales and references.

Line79ff: "Figure 1. Study sites. (a) Global distribution of salt marshes (purple), mangroves (green) and coexistence of both (blue) relative to the M2-tidal range^{20,21}, S1-6 and M1-6 are salt marsh and mangrove systems investigated in this study; (b) Example of distinct channel networks in salt marsh and mangrove systems. (c) Snapshot of different colonization strategies between salt marsh and mangrove systems. Salt marshes colonize slowly (~years²²) creating a patchy spatial pattern, whereas mangroves colonize fast(~weeks²⁵) creating a homogenous spatial pattern."

L82: 'Seems decoupled from tidal range' – these references don't support this point explicitly. Are there references that do, or is this decoupling something that this study needs to explicitly demonstrate itself, in which case this sentence will need rewriting to reflect. This touches on my point 2 above.

We agree with the reviewers point, adjusted the reference and clarified the text:

Line91ff: “Despite these shared generic feedbacks, observations from satellite imagery indicate striking differences in channel networks (abundance, size and extent) between mangroves and salt marshes across different systems (Fig. 1).”

L94: ‘Primary colonizers’ – what evidence is there for this ubiquity? There needs to be a reference to support this statement, and I don’t think those provided really do. Without a robust justification for focusing on *Spartina* (because it is almost everywhere), with its well-documented patchy colonisation patterns, the findings of the study could be quite niche, and only applicable to a few locations. I don’t question that *Spartina* and its growth forms are a reasonable basis for this study, but would like to see the global relevance better supported.

We added a reference showing the global distribution of the genus *Spartina* and moreover show its ubiquity in the Supplementary Figure.2a and 2b, that the genus *Spartina* is present in pioneer zones around the world.

Line 104: “The majority of temperate salt marshes host sub-species of the genus *Spartina* as their primary colonizers, which are characterized by relative low establishment probabilities from seeds but high rates of lateral clonal expansion leading to patchy vegetation cover during colonization^{33,34, 35} (Supplementary Fig. 2). ”

35. Moberley, D. G. (1953). Taxonomy and distribution of the genus *Spartina*. Iowa State University.

L95: ‘lateral expansion’ – do the authors mean ‘clonal expansion’?
Changed into: “lateral clonal expansion”

L103: some, but not all, plants in both systems produce a high density of cylinders.
We agree with the reviewer’s assessment, that stem density varies across salt marsh and mangrove species. However, plants of both systems are able to produce a high density arrays of cylindrical elements significantly altering flow and sediment transport condition within the plant structures compared to adjacent bare areas.

L106: assumption that network structure imprinted and stable... OK, this is well documented, but what about things like headward creek incision and so forth that is likely to lead to the densest channel networks? What about marshes in degenerative states where the initial channel network starts to become lost to internal dissection processes, which will dramatically alter the channel metrics.

We agree with the reviewer’s assessment that assuming the imprinted network structure to remain stable is well documented. Regarding headward creek incision which in mature salt marshes has been linked to the role of burrowing fauna creating favorable conditions for creek initiation and extension (e.g., Escapa et al., 2007, Hughes et al., 2009), we think that this can alter the previously imprinted network structure and will be part of the scatter observed throughout our field sites. However many authors have observed the relative stability of tidal creek systems once established (Ashley and Zeff, 1988; Novakowski et al., 2004) even where marginal erosion of creek banks is measurable (Gabet, 1998).

Escapa, M., Minkoff, D.R., Perillo, G.M., Iribarne, O., 2007. Direct and indirect effects of burrowing crab *Chasmagnathus granulatus* activities on erosion of southwest Atlantic *Sarcocornia*-dominated marshes. *Limnol. Oceanogr.* 52 (6), 2340–2349.

Hughes, Z.J., FitzGerald, D.M., Wilson, C.A., Pennings, S.C., Więski, K., Mahadevan, A., 2009. Rapid headward erosion of marsh creeks in response to relative sea level rise. *Geophys. Res. Lett.* 36 (3).

Ashley, G.M., Zeff, M.L., 1988. Tidal channel classification for a low-mesotidal salt marsh. *Mar. Geol.* 82 (1–2), 17–32.

Novakowski, K.I., Torres, R., Gardner, L.R., Voulgaris, G., 2004. Geomorphic analysis of tidal creek networks. *Water Resour. Res.* 40 (5), W05401.

Gabet, E.J., 1998. Lateral migration and bank erosion in a saltmarsh tidal channel in San Francisco Bay, California. *Estuar. Coast* 21 (4), 745–753.

Figure 2: There appears to me to be evidence within classes that the channel metrics are sensitive to tidal range, even if the differences between classes exceed these. See point 1 above.

See above, we agree with the reviewer's assessment that although vegetation type explains the main differences in the observed metrics, there may also be an effect of tidal range on the developed channel networks visible in Supplementary Figure 3.

We now added this to the text:

Line 295: "Our results underline that bio-geomorphological feedbacks acting on the watershed scale are able to form vastly different channel networks, with some scatter in network properties caused by . tidal range and relative elevation (Supplementary Table 1, Fig.3)."

Supplementary Figure 3: The authors claim linear relationships, but it looks like one could just as easily fit non-linear functions to the points presented. Can the authors provide fit statistics for linear and non-linear fits to show that the data do indeed best fit a linear relationship? Also spelling of Length on X axis label.

Figure 3 shows a power-law relationship in the shape of $L = a \text{Area}^b$ following previous examples of the Venice lagoon (e.g., Marani et al. 2003). We now also provide the fit-statistics in the figure caption.

Figure 4: Was the lab experiment only run once? If not, how many replicates, and can any statistical test be offered to show that the differences are significant? Admittedly, the contrast shown is convincing if only one test was conducted, but must then be interpreted more explicitly in the context of a single observation and qualitative comparison. The colours in the top-down photos don't reproduce brilliantly – can they be altered to make the vegetation elements more distinct from the elevation change overlay?

Since seedlings needed to be germinated and grown in order to run the lab-experiment, experiments were carried out once without additional replicates. However, pre-experiments were conducted with artificial obstructions which showed that the test is repeatable. We would also like to further refer to comparable experiments using the same approach (e.g. Stefanon et al. 2010, Kleinhans et al. 2015), which we believe lead to quantitative results. We thank the reviewer for the suggestion regarding the visibility of the vegetation, which we now adapted in the revised version.

Stefanon, Luana, et al. "Experimental analysis of tidal network growth and development." *Continental Shelf Research* 30.8 (2010): 950-962.

Kleinhans, Maarten G., et al. "Turning the tide: Growth and dynamics of a tidal basin and inlet in experiments." *Journal of Geophysical Research: Earth Surface* 120.1 (2015): 95-119.

L222: 'geometrically'
Changed as suggested

L256-262: Interesting questions that definitely need to be addressed. Is a major contrast between the sites also the sediment delivery rate?

Although this is an intriguing question, unfortunately we do not have data on sediment supply available for all the field sites. However, the geomorphological settings of our field sites point to erosional channel development. Thus we expect that differences in sediment supply will not alter our conclusions.

L263: The spatial scale at which plant colonisation is important compared to geological constrains, and whether there is anything in between is interesting. It also becomes a temporal scale issue as older networks are likely to be larger (even if inherited from smaller, plant-driven geometries). Larger networks also more easily detected at a given resolution... Do the authors get a sense of what they mean by 'local scales' and what the upper scale limit at which plant colonization effects are dominant might be? It seems possible that this upper limit might be different depending on the plant too. There are all sorts of interesting questions here!

We thank the reviewer for the enthusiasm regarding the interpretation of our results, which we share. The current study mainly focuses on wetland scale channel interactions and is unfortunately not able to distinguish between local/wetland scale and large/estuary scale effects. However, we would expect that the absence/presence of channels on the wetland scale might also be able to shape channel systems on the estuary/lagoon scale or at least control estuary/lagoon scale dynamics, for example through the alteration of sediment budgets.

L273: Please provide a reference for the 'deeper' argument.
Changed as suggested

D'Alpaos, A., Lanzoni, S., Mudd, S. M., & Fagherazzi, S. (2006). Modeling the influence of hydroperiod and vegetation on the cross-sectional formation of tidal channels. *Estuarine, Coastal and Shelf Science*, 69(3-4), 311-324.

L275: If channel planforms are more limited in mangroves, doesn't that imply that they will probably have to be more hydraulically efficient in order to exchange a given volume of sediment? This is obviously beyond the scope of this study, but doesn't hydraulic efficiency complicate the inferences made here about the implications of different networks for stability, surge attenuation and so forth?

We want to thank the reviewers for raising this aspect of our study. We would indeed expect that mangroves need to be more hydraulically efficient to exchange a given volume of water, nutrients and even sediments. However, following the definition of hydraulic resistance given in Persson et al. 1999 we do not expect them to affect our suggestion on wave attenuation, since it mainly focuses on residence time in wetlands rather than loss in momentum and energy through propagation through channels or vegetation structures.

Persson, Jesper, N. L. G. Somes, and T. H. F. Wong. "Hydraulics efficiency of constructed wetlands and ponds." *Water science and technology* 40.3 (1999): 291-300.

L279: 'determinants OF whether'
Changed as suggested

L281: 'effectiveLY'
Changed as suggested

Reviewers' Comments:

Reviewer #1:

Remarks to the Author:

The main changes that resulted from the first round of reviews consisted in the inclusion of an analysis of two field sites to ground-truth drainage networks obtained via remote sensing. This ground-truth was extremely important due to the potential limitations of remote sensing products to provide accurate drainage network properties, which may also be different in saltmarsh and mangrove. The authors compared satellite-based channel networks with high-resolution digital elevation models for one mangrove (Whitianga, New Zealand) and one salt marsh system (Saeftinghe, the Netherlands). I believe the comparison is valuable, but I am not sure that the authors have used the best way to analyse the DEMs and to compare them with the satellite data. I would have expected maps with the two networks where the differences could be clearly highlighted. I am also not sure I fully understand supplementary figure 9, but it looks like the corrections necessary to match the satellite network to the DEM are important. In fact, the new figure 3 that includes the corrections is quite different from the previous figure 3.

My concern is that the corrections are important and are based on one site for mangrove and one site for saltmarsh. What would have happened if different correction factors had been obtained from other sites? Would it be possible that correction factors from other ground-truthing result in rejection of the hypothesis of the authors (i.e. saltmarsh create more extensive channel network than mangroves)? I think that is a real possibility, as looking at the new figure 3 it can be seen that the difference between drainage networks of saltmarsh and mangrove is not as important as in the original figure 3. The authors should try harder to convince the reader that the corrections are reliable.

Figure 3 also shows that the Venice data (Marani et al., 2003) plot way above the general saltmarsh trend, but they are part of the data set. How was the Venice data (Marani et al., 2003) obtained? DEM or satellite?

Figure 2 shows that Krabbenkreek pushes the mean value of the saltmarsh ensemble way up. Because of the small sample size of the study, big changes in the general trend can happen if one site is removed from the analysis. The authors should discuss the robustness of their results in more detail. A minor point: Supplementary Figures 4 and 5 and Table 2 have not been updated to include the new sites

Reviewer #2:

Remarks to the Author:

I have read in detail and with great interest the revised version of the paper "Salt marshes create more extensive channel networks than mangroves" by Schwarz et al.

I consider that the Authors have quite satisfactorily covered, both by correcting or answering correctly, all the questions and comments that were posed by myself in the review of the first version. I have also read the response to the comments and requests for change of the other reviewers and, in my view, the paper greatly benefited from those suggestions and from the detailed answers to these reviewers. I am quite pleased to note that the authors have very seriously addressed all of the comments: they have done a thorough job in revising this manuscript. The paper has been improved and has advanced notably in clarity. I think this new version of the manuscript is scientifically stronger.

I think this paper will be of great interest to the readership of Nature Communications and I, therefore, recommend publication.

Sincerely,
Andrea D'Alpaos

Reviewer #4:

Remarks to the Author:

Having reviewed the revised manuscript and responses to reviewer comments I am satisfied that the authors have adequately addressed my concerns. I do, however, feel that they could have been more explicit in their textual amendments where they acknowledge limitations and assumptions identified by the reviewers.

REVIEWERS' COMMENTS

Reviewer #1 (Remarks to the Author):

The main changes that resulted from the first round of reviews consisted in the inclusion of an analysis of two field sites to ground-truth drainage networks obtained via remote sensing. This ground-truth was extremely important due to the potential limitations of remote sensing products to provide accurate drainage network properties, which may also be different in saltmarsh and mangrove. The authors compared satellite-based channel networks with high-resolution digital elevation models for one mangrove (Whitianga, New Zealand) and one salt marsh system (Saeftinghe, the Netherlands).

I believe the comparison is valuable, but I am not sure that the authors have used the best way to analyse the DEMs and to compare them with the satellite data. I would have expected maps with the two networks where the differences could be clearly highlighted.

We thank the reviewer for recognizing the value of the additional analysis for robustness of our results. Regarding the comparison between channel networks extracted from the DEMs and the satellite data, the method of removing channel of different Hortonian-orders was chosen to be able to remove artefacts caused by the limited resolution in satellite products. For the sake of completeness we now also added the mangrove channel network extracted from digital elevation models (new Supplementary Figure 9b and underline the validity of the corrections-factors in Fig. 9d,e)

I am also not sure I fully understand supplementary figure 9, but it looks like the corrections necessary to match the satellite network to the DEM are important. In fact, the new figure 3 that includes the corrections is quite different from the previous figure 3.

We thank the reviewer for pointing this out and now added more detailed explanations to the sensitivity tests and Fig.9 in the supplementary material and the method sections:

Line 353: “Channel network analysis based on satellite data can be complicated by the limits in available spatial resolution (Planet Labs cell-size: 3 m) and by above-ground biomass (e.g. salt marsh grasses or mangrove crowns) of wetland vegetation obscuring small (low Hortonian order) channels or creeks. Therefore, we carried out a sensitivity analysis, establishing correction factors to account for these inaccuracies based on the following steps. (1) We compared channel networks extracted from satellite data with channel networks extracted from high-resolution digital elevation models for one mangrove (Whitianga, New Zealand)⁶⁶ and one salt marsh system (Saeftinghe, the Netherlands) using the same extraction procedure (Rijkswaterstaat, RWS.nl). This comparison resulted in visible differences between satellite and DEM based channel metrics. (2) To correct for these differences in our global data set we successively removed lower order channels (1st - 5th order) of the DEM data and compared extracted channel metrics with that of the satellite data (Supplementary Fig. 9a,b). (3) We then established an exponential relationship between removed channel order and channel metrics for mangroves and salt marshes, which was used to correct the satellite based metrics (Supplementary Fig. 9c).”

My concern is that the corrections are important and are based on one site for mangrove and one

site for saltmarsh. What would have happened if different correction factors had been obtained from other sites? Would it be possible that correction factors from other ground-truthing result in rejection of the hypothesis of the authors (i.e. saltmarsh create more extensive channel network than mangroves)?

We thank the reviewer for pointing this out. We agree that additional reference sites with high resolution elevation data would be desirable. However, these data (especially for mangrove systems) is unfortunately extremely scarce, which is why the presented methodology based on satellite imagery was developed. We agree that the correction factor should be further explored, and this highlights the need to obtain new DEMs from a broader range of vegetated coastal landscapes. Still, both before and after the correction factor, our analysis (Fig.2) shows that the extracted metrics for mangroves and salt marshes vary by an order of magnitude. We therefore expect that improved correction factors could nuance our results but will not change the presented trends nor jeopardize the general conclusion.

We now add that in:

Line 367: “Due to the scarcity of high resolution elevation data, correction factors were based on only one system per wetland type.”

I think that is a real possibility, as looking at the new figure 3 it can be seen that the difference between drainage networks of saltmarsh and mangrove is not as important as in the original figure 3. The authors should try harder to convince the reader that the corrections are reliable. Figure 3 also shows that the Venice data (Marani et al., 2003) plot way above the general saltmarsh trend, but they are part of the data set. How was the Venice data (Marani et al., 2003) obtained? DEM or satellite? Figure 2 shows that Krabbenkreek pushes the mean value of the saltmarsh ensemble way up. Because of the small sample size of the study, big changes in the general trend can happen if one site is removed from the analysis. The authors should discuss the robustness of their results in more detail.

We thank the reviewer for this comment. Indeed, the data of Marani et al. (2003) are based on digital terrain maps which means, if our corrections are working, the salt marsh data should at least partially overlap with the relationship proposed by Marani et al. (2003). We re-examined the data presented in Fig. 3 and realized that although corrections were applied to mUPL and DD (Fig. 2), they were not applied to channel lengths (Fig. 3). We have now corrected this (more information can be found in the supplementary material, Fig 9d,e, which also shows the validity of our correction approach). With the applied channel length correction, indeed our data approach the scaling found by Marani et al. (2003), which supports the used correction factor.

This was also added into the main text:

Line 213: “More information on the influence of the correction factor on this relationship can be found in Supplementary Fig.9d,e.”

Additionally we extended our explanation of the results of new Fig.3:

Line 182ff:

“A comparison between the total channel length, the sum of all channel lengths, as a function of the watershed area follows the previously described tidal range independent power law relationship found for salt marshes in Great Britain, Italy and the US^{43,46,49} (Fig.3). The increased slope (b) for salt marshes compared to mangroves could hint to a faster increase in channel length to drainage area. However the 5 green mangrove points on the left side of the graph (Area: $10^3 - 10^4$) driving the difference in this relationship belong to a very small mangrove system (Whitianga) which might offset the otherwise spatially constant scaling of network development⁴⁶, or suggest that power law scaling is less pronounced at mangrove systems (Fig.3).”

Moreover, we tested the impact of Krabbenkreek, resulting in a significant relationship, even if this system would be removed.

A minor point: Supplementary Figures 4 and 5 and Table 2 have not been updated to include the new sites

We added the information as suggested.

Reviewer #2 (Remarks to the Author):

I have read in detail and with great interest the revised version of the paper “Salt marshes create more extensive channel networks than mangroves” by Schwarz et al.

I consider that the Authors have quite satisfactorily covered, both by correcting or answering correctly, all the questions and comments that were posed by myself in the review of the first version. I have also read the response to the comments and requests for change of the other reviewers and, in my view, the paper greatly benefited from those suggestions and from the detailed answers to these reviewers. I am quite pleased to note that the authors have very seriously addressed all of the comments: they have done a thorough job in revising this manuscript. The paper has been improved and has advanced notably in clarity. I think this new version of the manuscript is scientifically stronger.

I think this paper will be of great interest to the readership of Nature Communications and I, therefore, recommend publication.

Sincerely,
Andrea D'Alpaos

We thank the reviewer for the positive assessment of our study.

Reviewer #4 (Remarks to the Author):

Having reviewed the revised manuscript and responses to reviewer comments I am satisfied that the authors have adequately addressed my concerns. I do, however, feel that they could have been more explicit in their textual amendments where they acknowledge limitations and assumptions identified by the reviewers.

We thank the reviewer for the positive assessment of our study. As mentioned above in response to the comment by reviewer 1, we have now added more detailed explanations to the sensitivity tests and Fig.9 in the supplementary material, and added the following text in the method section of the manuscript:

Line 353: “Channel network analysis based on satellite data can be complicated by the limits in available spatial resolution (Planet Labs cell-size: 3 m) and by above-ground biomass (e.g. salt marsh grasses or mangrove crowns) of wetland vegetation obscuring small (low Hortonian order) channels or creeks. Therefore, we carried out a sensitivity analysis, establishing correction factors to account for these inaccuracies based on the following steps. (1) We compared channel networks extracted from satellite data with channel networks extracted from high-resolution digital elevation models for one mangrove (Whitianga, New Zealand)⁶⁶ and one salt marsh system (Saefthinghe, the Netherlands) using the same extraction procedure (Rijkswaterstaat, RWS.nl). This comparison resulted in visible differences between satellite and DEM based channel metrics. (2) To correct for these differences in our global data set we successively removed lower order channels (1st - 5th order) of the DEM data and compared extracted channel metrics with that of the satellite data (Supplementary Fig. 9a,b). (3) We then established an exponential relationship between removed channel order and channel metrics for mangroves and salt marshes, which was used to correct the satellite based metrics (Supplementary Fig. 9c).”